# In Situ Tungsten Carbide Formation in Nanostructured Copper Matrix Composite Using Mechanical Alloying and Sintering

**DOI:** 10.3390/ma15072340

**Published:** 2022-03-22

**Authors:** Mahani Yusoff, Hussain Zuhailawati

**Affiliations:** 1Faculty of Bioengineering and Technology, Universiti Malaysia Kelantan, Jeli 17600, Kelantan, Malaysia; 2School of Materials and Mineral Resources Engineering, Engineering Campus, Universiti Sains Malaysia, Nibong Tebal 14300, Penang, Malaysia; zuhaila@usm.my

**Keywords:** tungsten carbide, in situ formation, copper matrix composite, mechanical alloying

## Abstract

In this study, an in situ nanostructured copper tungsten carbide composite was synthesized by mechanical alloying (MA) and the powder metallurgy route. The microstructure and phase changes of the composite were characterized by X-ray diffraction, scanning electron microscopy and X-ray photoelectron spectroscopy. Tungsten carbide phases (WC and W_2_C) were only present after MA and combination of sintering. Higher energy associated with a longer milling time was beneficial for the formation of WC. Formation of W_2_C and WC resulted from internal refinement due to heavy plastic deformation in the composite. The solubility of the phases in the as-milled and sintered composite was described by the changes of the lattice parameter of Cu. Chemical analysis of the surface of a composite of W 4f and C 1s revealed that the increased defects introduced by MA affect the atomic binding of the W-C interaction.

## 1. Introduction

Copper (Cu) possesses excellent resistance towards corrosion and oxidation. It is more favorable among other common metals in electrical applications because of its excellent electrical and thermal conductivity. Despite the fact that Cu resistivity (1.7 μΩcm) is similar to that of silver resistivity (1.6 μΩcm), Cu has historically been used as a high temperature conductor (700–800 °C). There has been much attention in recent years focusing on copper-based composites reinforced with transition metal carbides, borides and nitrides. These ceramic reinforcements tend to show metal-like bonding with Cu as long as heat formation does not become more negative. Furthermore, the presence of carbide may retard particle coarsening during annealing in order to retain the strength of the composite. In electro-friction application of contact materials, carbide was purposely added into Cu to increase mechanical properties [1,2,3,4,5,6,7,8,9,10] and wear resistance [11,12,13,14,15,16]. Tungsten carbide has unique properties, such as high melting point, extremely high hardness, fracture toughness, compressive strength and excellent corrosion resistance and thermal stability. Tungsten carbide has lower density of about 15.63 g/cm^3^ over tungsten that is 19.3 g/cm^3^ [17]. Tungsten carbide is the hardest carbide at elevated temperatures (1000 kg/mm^2^) at 1000 °C [18] and also retains its stability when heated and cooled [17]. These unique properties make tungsten carbide suitable to reinforce soft copper for mechanical properties improvement. 

In situ metal matrix composite can be prepared through formation of solid solution by subjecting the elemental mixture powder to a highly energetic milling or mechanical alloying (MA). MA has been known as a powerful technique to synthesis a variety of materials, including immiscible and miscible systems. The ability of MA to disperse very fine reinforced particle into copper leading to a better control of microstructure makes it the most promising method in the copper strengthening process. The main characteristics of MA are its roles in increasing the grain boundary volume and dislocation density. A large amount of energy stored alongside the grain boundaries accompanied by very high density defects, causes the formation of metastable or stable phases to occur at lower temperatures. In equilibrium condition, the maximum energy at the grain boundary can reach as high 10 kJ/mol with a grain size of 1 nm and a dislocation density of 1016 mm^−3^ [19]. This energy is then used to promote dissolution among the phases in the powder mixture. Therefore, nanostructured material can be produced by either creating new solid solution phase or forming a supersaturated solution. Besides, it has been reported that nanocomposite with grain size ranging from 5 to 50 nm [20,21,22,23,24,25] and dislocation densities around 1015–1016 mm^−3^ can be obtained by MA [26]. Moreover, a recently developed method combining both powder metallurgy (PM) and MA has produced very excellent results on the mechanical properties of various Cu-based composite systems [27,28,29,30,31]. 

The most stable tungsten carbide is WC, which has a hexagonal structure with two atoms per unit *a* with *c/a* ratio of 0.976. However, due to the presence of semi carbide (W_2_C), it is practically impossible to synthesize only WC single-carbide in copper composite. W_2_C is a sub-carbide with a hexagonal structure and lattice parameters *a* = 0.38 nm and *c* = 0.47 nm. In addition, few researchers have addressed the microstructural and structural evolution on the behavior of in situ tungsten carbide phase formation in copper matrices synthesized by MA and PM in the literature [32]. Therefore, the main purpose of this study is to describe the formation and distribution of in situ formed tungsten carbide in nanostructured copper-based composite. The effect of milling time on the microstructural and structural of the composite is also discussed. 

## 2. Materials and Methods

Copper (22.3 μm), tungsten (11.4 μm), and graphite (17.0 μm) powder mixtures were dry milled in a planetary ball mill (Fristch Pulverisette 5). The ball-to-powder ratio (10:1) was kept constant throughout the milling process using 10 mm balls and n-heptane as a process control agent. The milling process was carried out using a stainless-steel milling jar and balls (Fristch, Markt Einersheim, Germany). The powder mixture was loaded in the vial sealed with a rubber ‘O-ring’ and filled with argon gas. MA was performed for several milling durations (10, 20, 40 and 60 h) without interruption with the rotation speed of 400 rpm. The as-milled powder was then cold compacted in a 10 mm diameter die under pressure of 300 MPa. Sintering (Lenton LTF 14 tube furnace, Nottingham, UK) was performed in an argon environment at 900 °C with heating rate of 5 °C/min for one hour.

Phase identification for both milling and sintered products was conducted by X-ray diffraction analysis (XRD) using a Bruker AXS D8 Advance (Fitchburg, WI, USA) using Cu Kα radiation (wavelength, λ = 0.154 nm) at room temperature with the scan range 20–80 °C of 2θ angle. The well-known Williamson–Hall method was used to determine the crystallite size and internal strain of the as-milled powders. The changes of binding energy for Cu, W and carbon after mechanical alloying time were measured by X-ray photoelectron spectroscopy (XPS). To achieve clean surfaces, the samples were evacuated under argon in maintained ultra-high vacuum of 3 × 10^−9^ Torr. The scanning process was performed by pass energy of 160 eV for each specimen and core level scan were operated at pass energy of 20 eV with step size of 0.1 eV. Microstructure investigation was performed by field emission scanning electron microscopy (FESEM, Supra 55VP, Carl Zeiss Microscopy GmbH, Oberkochen, Germany) at 30 kV equipped with energy dispersive X-ray (EDX, Oxford. Instruments, Abingdon, UK) analysis.

## 3. Results and Discussion

### 3.1. Cu-W-C as-Milled Powder

Figure 1 shows the XRD patterns for the as-milled powders with different milling times. Only two crystalline phases were obtained with no sign of the appearance of the amorphous phase. Graphite has not been observed for all milling times as a result of its faster diffusion than W in Cu matrix. The reason is that solid solubility for graphite in Cu is far behind that of W, as can later be converted into solid Cu(C) solution. Meanwhile, Cu and W are immiscible to each other at room temperature; therefore, to have a maximum solubility is almost impossible. Moreover, peaks of Cu and W broadened with a little shift to the left (red line) with the prolonged milling time up to 60 h. 

The broadening of the peak represents structural refinement that occurred in the as-milled powders. In this work, calculation of crystallite size and internal strain by Williamson–Hall only uses two high density peaks of Cu ((111) and (200) reflections) since another peak at higher angle tends to broaden and overlap with the W peak. Estimation of the crystallite size and internal strain has been done after removing the instrument broadening by reference sample of external Si standard. As shown in Figure 2, Cu crystallite size reduced to below 16 nm with increased milling time. As the powder mixtures undergo MA, plastic deformation causes Cu crystallite to slip passing through each other which contributes to the change of its size. At higher supplied energy to the powders, each crystallite remains constrained by its surrounding crystallites and produced stress/strain in the composite. Higher milling time should have lower Cu crystallite size due to the fact that repeated deformation occurs at a higher rate. Besides, decreases of crystallite size depend very much on the accumulation of defects. In most cases of MA product, the heavily stressed area is mainly caused by the increase in dislocation densities (Figure 3). Hence, Cu internal strain behaves inversely to that crystallite size for all milling times.

The solubility activity in the Cu-W-C mixture can be described by the variation of Cu lattice parameter against milling time, as shown in Figure 4. The lattice parameter was calculated using Cohen’s method for a cubic structure [33]. The plot clearly indicates that increases milling time increases the Cu matrix lattice parameter. For pure metals, milling duration has a minor effect on lattice changes [34,35]. Hence, the expansion of Cu matrix lattice from 0.363 nm (pure copper) was attribute to the formation of Cu(C) solution and Cu-W mixture. However, solute graphite in Cu matrix did not yield any significant effect because of its strictly Van der Waals interactions [36]. Due to the small difference in atomic radius between Cu and W atoms (0.138 and 0.128 nm, respectively), simultaneous atom substitution occurs, which is consistent with Vegard’s law’s principle. According to Vegard’s law, the lattice parameter of a two-phase solid solution is linearly related to the solute concentration. This variation is consistent with lattice change and imperfection caused by continuous particle deformation.

Figure 5 shows the SEM micrographs of as-milled Cu-W-C mixture after various milling times. Two kinds of particles can be obtained. The white particles represent rich sides of W whereas the grey particles are rich sides of Cu confirmed by EDX analysis in Figure 6. The as-milled powders in this figure show uniform particle distribution in all milling periods associated with small particle size and shape. In the beginning of milling (10 h), Cu particles were welded, whereas irregular W particles were embedded alongside the soft and ductile welded Cu. These particles became finer when milling time was increased owing to the beginning of particle fracturing as the milling reached 20 h and 40 h. Further milling to 60 h may heavily fracture the W particles with the conversion from irregular into spherical shape with average particle size of 1 μm.

### 3.2. Sintered Cu-W-C Composite

XRD patterns of sintered composite are shown in Figure 7 which indicates the diffraction lines of W gradually disappeared with increasing milling time. WC and W_2_C were traced at even lower milling time and there are no changes in the phases obtained, but the peak of W_2_C is reduced for 10 h to 40 h of milling. Between these two phases, W_2_C is suggested to form first, before WC, in less than 10 h of milling. The fact that W_2_C was first observed at lower milling time is because it is less thermodynamically stable than WC. WC and W_2_C have nearly equal stabilities at room temperature since heat formation of W_2_C only occurs at over 0.12 eV per carbon atom compared to WC (0.42 eV) [37]. However, W_2_C is found to be unstable at low temperature range from 677 to 1000 °C [38,39]. Furthermore, it is possible that lower milling has induced a lower amount of C to react with W than unreacted W [40,41,42]. W atoms have a greater number of W neighbors than in the W-C contact due to higher concentration of starting W. The reaction for the W_2_C formation can be suggested by Equation (1):(1)$$2W+C\to W_{2}C$$

Meanwhile, another important finding was that the heat supplied during sintering has successfully produced stable WC. This result indicates direct reaction of W+C→WC has started at an even shorter time. Obviously, when milling time was extended, W_2_C phase reduced while WC phase increased. According to Bolokang et al., 2010, [41] and Mehrizi et al., 2019 [42], WC formation attributed to the reaction of $W_{2}C+C\to 2\mathrm{WC}$ and W+C→WC are essentially obtained by higher available carbon distribution which can be produced at longer milling time. The extent of WC formation also behaves the same way after further milling. It may be that the reduced unreacted W and the disappearance of W_2_C peak at longer milling increased the WC phase. 

WC or W_2_C was not formed during milling but somehow observed after subsequent sintering at 900 °C. The energy for the formation might came by both milling and heating since mechanical energy obtained from milling alone is insufficient to overcome the activation energy of tungsten carbide formation. Heavy internal refinement provided by milling brought C atoms into intimate contact with W leading to formation of W-C composite particles. With repeated fracturing and rewelding, defects and internal strain with greater amounts of grain boundaries were generated. Therefore, C atoms can diffuse along W grain boundaries as a result of many diffusion pathways having been created during the milling process. The finding of this work is in agreement with Li et al., 2019 [32], who proved that in situ formation of WC can only be obtained after MA and additional heat treatment, respectively. Although longer milling might provide more energy for the tungsten carbide formation, it could be useless when considering the Fe contamination from the milling jar and balls. In this work, further milling to 60 h did not yield enough energy to be supplied to the powders for tungsten carbide formation, but instead produced contaminated Fe_3_W_3_C phase, a trend which is similar to the work of Zhong et al., 2019 [43].

Figure 8 demonstrates the crystallite size of Cu with increasing milling time. The internal strain calculation was unsuccessful since the slope in the Williamson–Hall plot is linear for all milling times. Therefore, stress-free composite was obtained after sintering but still maintained its nanostructure. The lowest Cu crystallite size of the sintered composite was obtained by 40 h of milling because of the presence of hard second phases. It seems that faster diffusion is promoted by smaller crystallite size [44]. Baghani et al., 2018 [24], describes that crystallite size decrease is originated by atomic size mismatch among the phases in the CuFe-Al_2_O_3_ nanocomposite, which is also applicable to the explanation for Cu-W-C composite. Therefore, the increased of Cu crystallite size at 60 h milling is suggested to be as a result of the growth of Cu grain. 

Results of XPS analysis on the composite prepared from powder milled for 40 h and the unmilled composite are shown in Figure 9. Spectra collected from wide scan were distinguished according to C 1s, O 1s, Cu 2p and W 4f. As can be seen, the XRD pattern of the composite milled at 40 h for the presence of tungsten carbide phase is well agreed and supported by XPS analysis. High intensity of graphite C 1s and W 4f peaks are determined in the composite milled for 40 h compared to that of the unmilled one, indicating that the MA process eventually altered the binding energy of the constituent elements.

Figure 10 shows the high resolution XPS spectra for the composite milled for 40 h. The atomic interaction in Cu matrix is described in Figure 10a. The appearance of –OH and –CH groups showed that PCA did not completely decompose upon extensive milling and sintering which accumulates very much on the surface of the composite. Clearly, a strong shake-up satellite for CuO is also noted in this figure. A reasonable fit to Cu metallic is assigned at 933.15 eV peak. Meanwhile, the atomic level scan for C 1s spectra is presented in Figure 10b and fits to four interactions are seen. The first peak of C 1s at 283.06 eV is attributed to the reaction of W and C. Spectrum of graphite is recognized at 284.33 eV (second peak) and shifted compared to the unmilled composite. The third and fourth peaks are ascribed as contaminated phase. On the other hand, Figure 10c shows two resolved peaks corresponding to the W 4f_5/2_ and W 4f_7/2_. However, only W 4f_7/2_ will further be discussed. The peak located at 31.35 eV corresponds to the metallic W and the peak at 32.19 eV resulted from the reaction between W and C, which can be proved by the change of binding energy of W 4f that is obviously related to the change of C 1s electron. The chemical shift of W 4f and C 1s in this composite is highly dependent on the rate of defects [45,46] formed during MA. The other higher peak at 32.75 eV explained that the composite was contaminated by oxygen that can be suggested to originate either from milling or sintering.

Figure 11 shows the SEM images of sintered composite at various milling times. Three areas can be distinguished in all milling times and consist of some non-uniformity of microstructure. The black area corresponds to porosities having various sizes and shape and the size reduced with increasing milling time. Very often, cold compaction leads to a porosity and void after sintering since there is a situation where no plastic deformation is involved during consolidation. The grey area consists of higher concentration of Cu and solution of Cu(C) which can be proved by elemental analysis in Figure 12. Tungsten carbide is more likely to solute in the Cu only at further milling. With the presence of tungsten carbide particles, the diffusion pathway of Cu particles is retarded, and much longer time may be needed to fracture into small size and distribute it in the Cu matrix. The regions identified in Figure 12 belong to higher accumulated of Cu where the other area was composed of Cu, W, Fe, O and C elements. Interparticle diffusion during sintering increased the homogeneity of microstructure, which can be shown by the area consisting of different color embedded in the red region (Cu) (Figure 12). Apart from that, there is no clear morphological difference between W_2_C and WC because WC was generated by transforming W_2_C, thus the microstructure should be identical.

## 4. Conclusions

Elemental Cu, W and C were used to synthesize nanostructured copper tungsten carbide. No carbide phase was obtained in the as-milled powder. It can only be formed after the sintering process. WC was formed with the coexistence of the reduction of W_2_C and unreacted W. Chemical shift of the composite proves that the composite contains W and C interaction that was promoted mainly by heavy deformation of MA. Progressive milling reduced the Cu crystallite size and increased the internal strain in the as-milled powder. The sintering process successfully relieved the stress in the composite while maintaining its nanostructure. 

## Figures and Tables

**Figure 1 materials-15-02340-f001:**
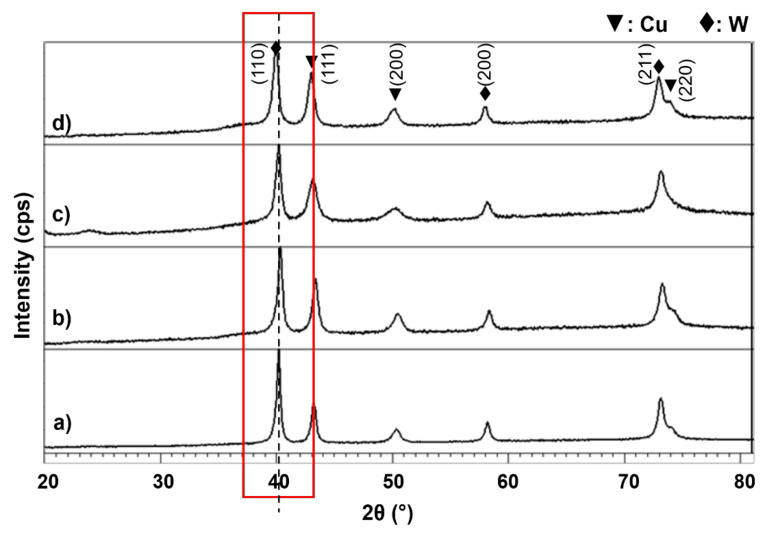
XRD patterns of the composite milled for (**a**) 10 h, (**b**) 20 h, (**c**) 40 h and (**d**) 60 h.

**Figure 2 materials-15-02340-f002:**
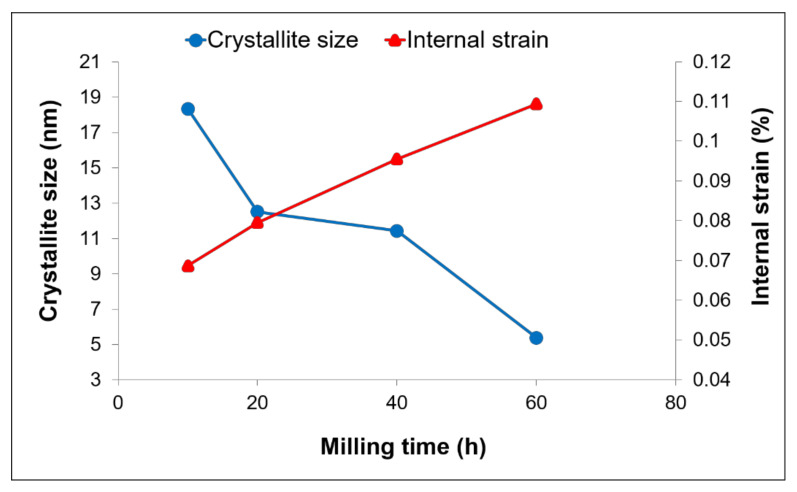
Cu crystallite size and internal strain of Cu-W-C powder for various milling time.

**Figure 3 materials-15-02340-f003:**
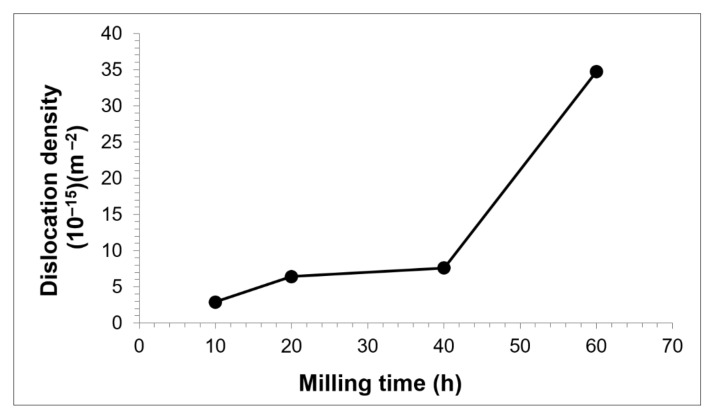
Dislocation density of as-milled Cu-W-C powder at various milling time.

**Figure 4 materials-15-02340-f004:**
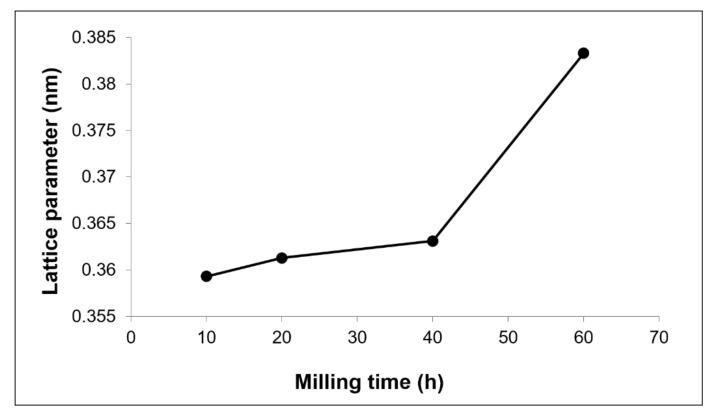
Cu lattice parameter of Cu-W-C powder at various milling time.

**Figure 5 materials-15-02340-f005:**
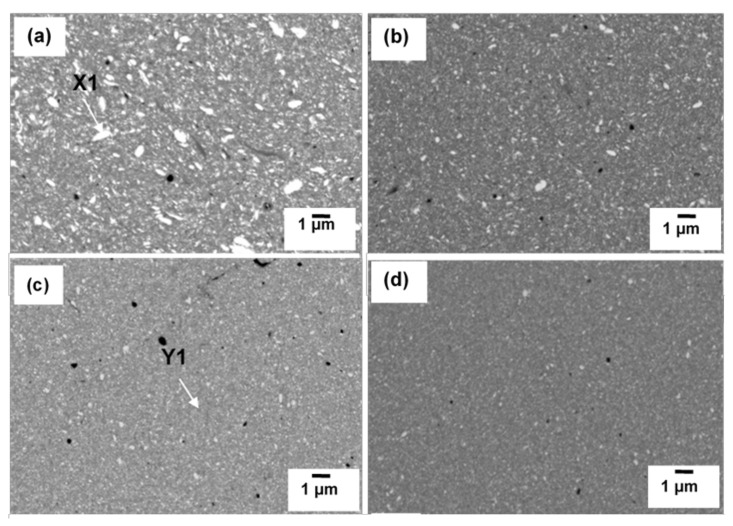
SEM images of Cu-W-C powder milled for (**a**) 10 h, (**b**) 20 h, (**c**) 40 h and (**d**) 60 h.

**Figure 6 materials-15-02340-f006:**
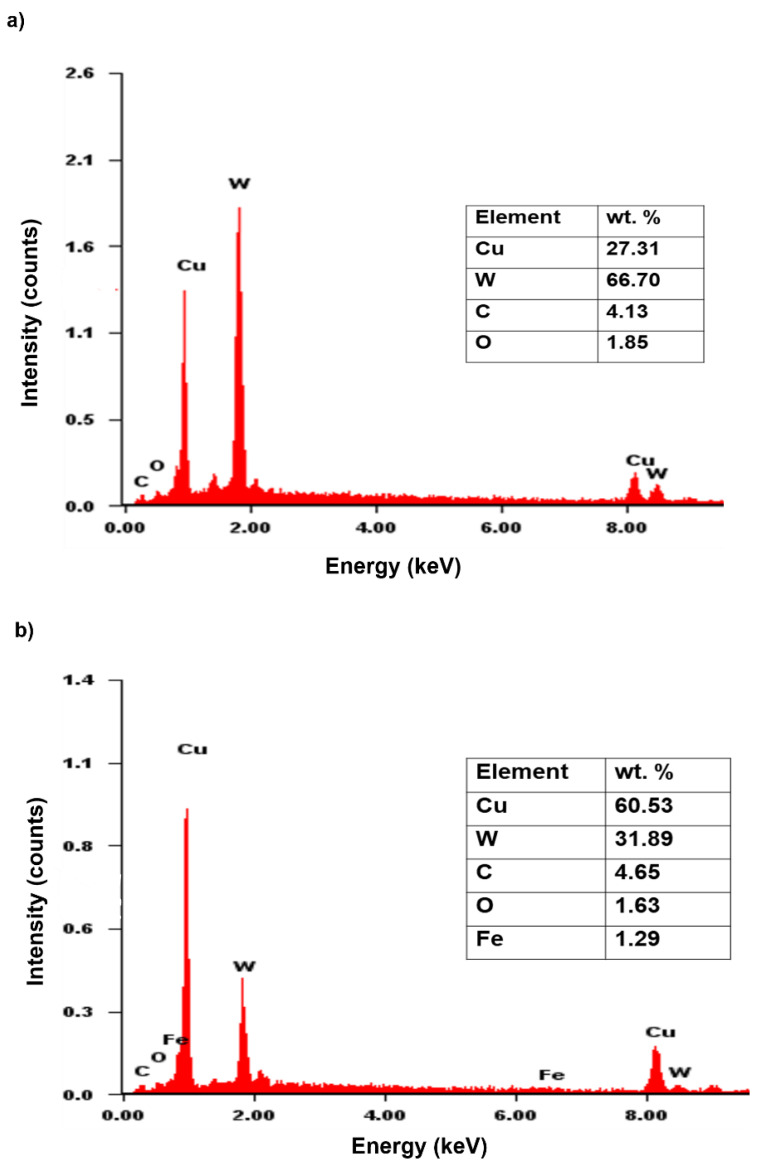
EDX analysis of (**a**) white particle (X1 area) and (**b**) grey particle (Y1 area) corresponding to Figure 5a,c, respectively.

**Figure 7 materials-15-02340-f007:**
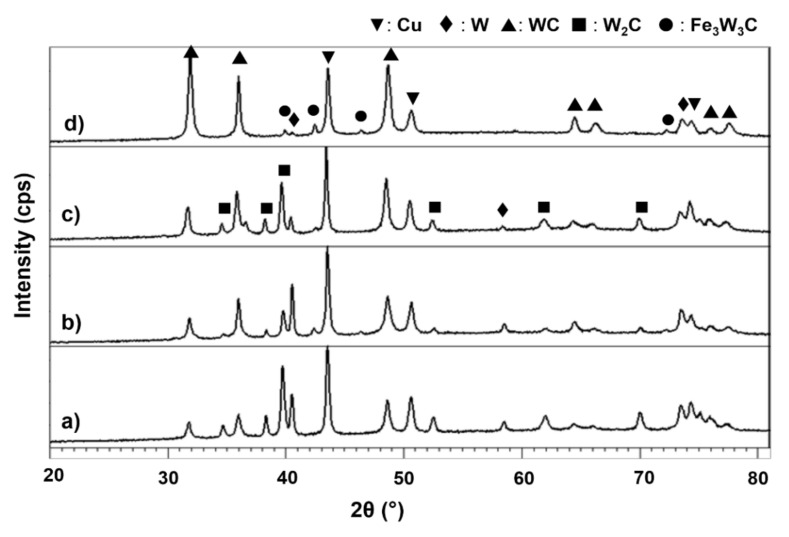
XRD patterns of Cu-W-C sintered composite milled for (**a**) 10 h (**b**) 20 h, (**c**) 40 h and (**d**) 60 h.

**Figure 8 materials-15-02340-f008:**
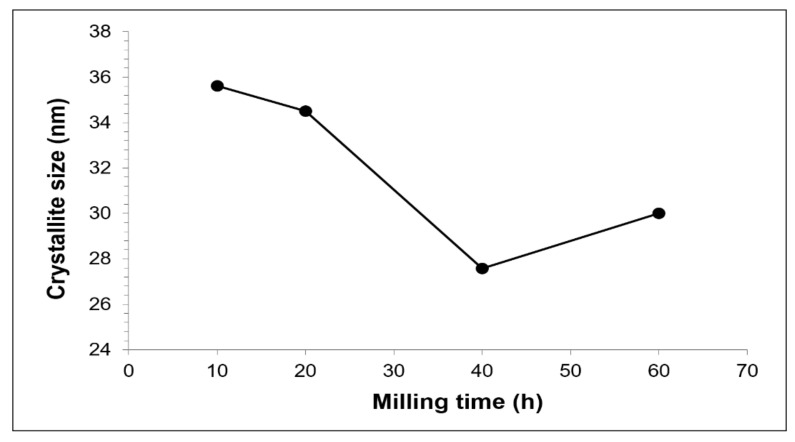
Cu crystallite size in Cu-W-C sintered composite at various milling times.

**Figure 9 materials-15-02340-f009:**
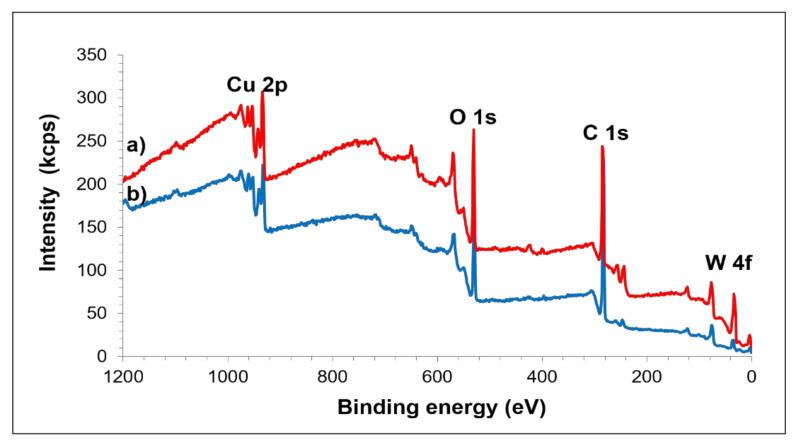
XPS wide scan spectra of the (**a**) sintered unmilled Cu-W-C mixture and (**b**) Cu-W-C sintered composite milled at 40 h.

**Figure 10 materials-15-02340-f010:**
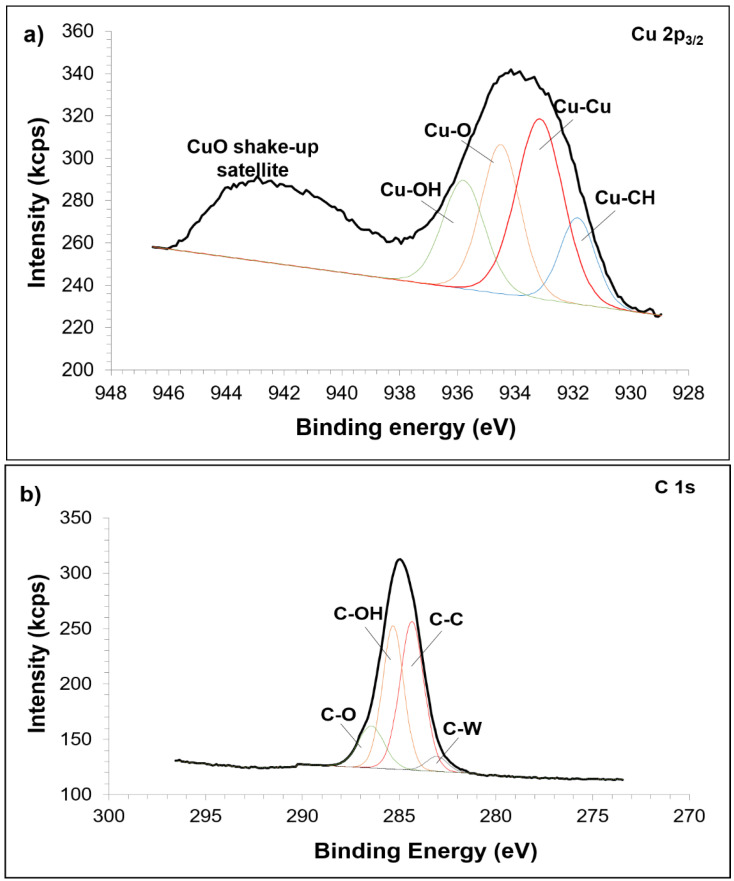
XPS curve fitting for (**a**) Cu 2p, (**b**) C 1s and (**c**) W 4f for sintered composite milled for 40 h.

**Figure 11 materials-15-02340-f011:**
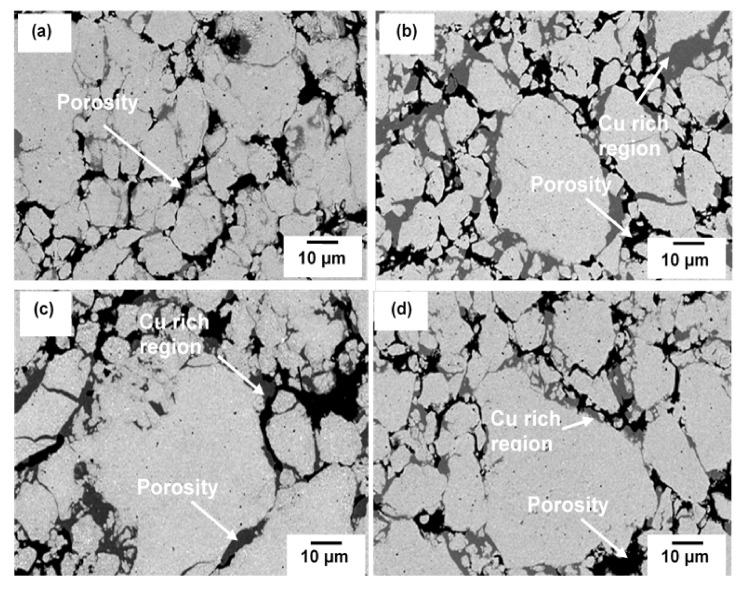
SEM images of Cu-W-C sintered composite milled at (**a**) 10 h, (**b**) 20 h, (**c**) 40 h and (**d**) 60 h.

**Figure 12 materials-15-02340-f012:**
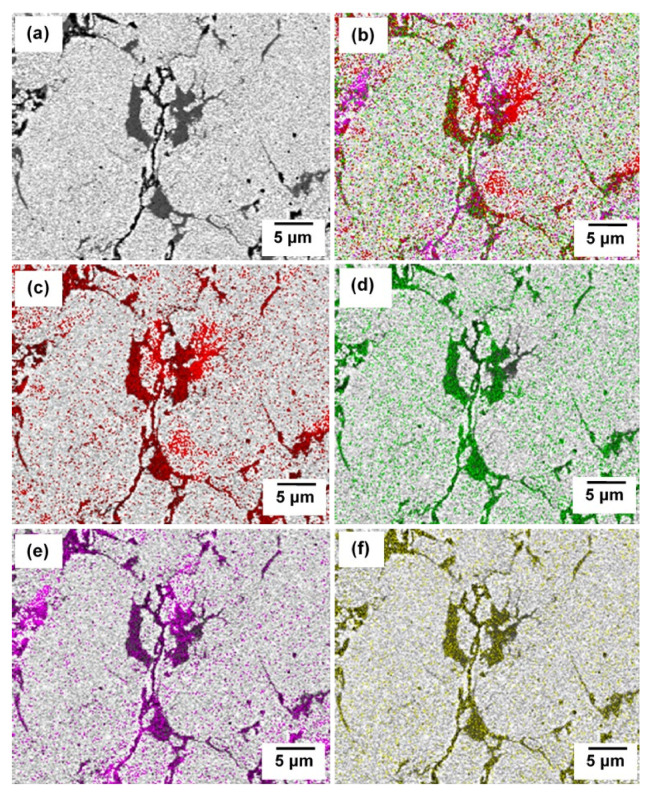
(**a**) SEM image and elemental mapping on the corresponding area of (**b**) all elements, (**c**) Cu (red), (**d**) W (green), (**e**) Fe (indigo) and (**f**) C (yellow) in sintered compact milled at 40 h and sintered at 900 °C.

## Data Availability

The data presented in this study are available on request from the corresponding author.

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
