# Peer review of "In Situ Tungsten Carbide Formation in Nanostructured Copper Matrix Composite Using Mechanical Alloying and Sintering"

_materials, 2022, doi:10.3390/ma15072340_

Round 1

Reviewer 1 Report

In this work, the authors focused on the study of the properties of the nanostructured copper-carbide composite of tungsten carbide.

The paper is very original and very interesting.

The authors adopted the appropriate research methodology.

Clear language is used.

The graphics and figures used in the paper are very clear.

 Interpretation of the tables is appropriate.

The authors drew proper conclusions.

This paper can be published.

Author Response

Thank you for reviewing our manuscript and thank you for comments and recommendations. 

Reviewer 2 Report

The content of the manuscript  requires further more attention of the authors to make it more attractive for publication acceptance. The report of the reviewer as proposed can be use as a serious tool of improvement.  
The manuscript content gives a good idea of the work undertaken which has some interest for the scientific community of composite materials based on metallic matrixes. It is actually shored up by sufficient references but the presentation and discussion of the different reported results can be improved. Despite the originality of the study announced in introduction, which is somehow shown in their proposal, I believe the authors will furthermore convince examiners if they carefully consider the remarks as below whose objective is a help for increasing the credibility of their work. 

I – General recommendations –
A) – The authors should have a look on their different manuscript sections to realize there is no separated “Discussion” part which seems probably as included under the “Results” heading. The two parts are often merged into a global section entitled “Results and Discussion”. If so here, redaction of their text should be more adapted. 
B) – Sometimes, the authors are greatly helpful to the reader duty when they use different colors to represent the plotting of different material characteristics in the same (2D) figure. This seems necessary here in the cases of Figs. (2), (8) and (9). 
II - On the scientific content of the submission – 
A) – In figure 1, the XRD peaks of the investigated materials look anonymous as reported. No [h k l] indices are attributed to each one of them while they suddenly appear in line 100 of the text. How to establish a clear relationship between the text and the reported results of the figure?? … 
B) – In the description of sub-section 3.1 (at line 90), the authors claim that “peaks of Cu and W broadened with a little shift to the left….”. These very important observations (broadening and shifting) are not at all evident in Fig.1. The authors are advised to propose first a global diagram pattern (from 20 < 2θ < 80) as done. Afterwards, they can focus their study in 
a limited range of 2θ values linked to a selected highly resolved peak of Cu or W element for a confirmation. 
C) – Nowhere in the text, the authors say something on the crystallographic structure of the investigated materials. Lattice parameter appears in line 123 and also on Fig.4 without any related indication. Before drawing a conclusion regarding the effects of milling time treatment on Cu-W-C composites, the authors should first be ensured that the lattice parameter of Cu matrix itself taken separately is quite indifferent to the same treatment. Some words on that point should be provided. 
 D) – In the comments of lines 152 and 153 linked to the results obtained from EDAX analysis, the authors referred to unexpected figs.4 (e)?? and (f)?? that do not exist in the manuscript. Obviously, they thought of figs.5 (e) and (f) proposed in page 5. For that purpose, please, increase the visibility of these two figures since reading on them is very difficult even with the help of corrective glasses. Besides, in the same fig.5, the reader wonders why two different SEM images [(c) and (d)] correspond to the same milling time (60h) while no image is provided for time 40h?? (see the description of line 161). 
E) – The comments of Fig.6 in sub-section 3.2. make the reader confused on the chemistry conventional rules is arisen here. All over sub-section 3.2, W2C and W2C are reported. Can the authors explain the difference between the two compounds?? 
G) – The comments of Fig.7 from line 209 till 217 are not fully in agreement with the results proposed. Actually, the presence of Fe2W2C compound as proposed in the reported XRD patterns is not at all expected since Fe element is not included is the investigated composite materials. Can the authors explain this fact?? Once again, [h k l] indices are particularly missing 
in the diffraction patterns and the discussion on the reported results is very poor while the effects of the milling time seem more pronounced here compared to the milled powder materials. Etc…. So many other remarks could be added from the remaining part of the manuscript. As it appears, a great deal of weaknesses in the manuscript are clearly revealed through the 
present report. The authors should be aware that a submitted paper cannot be considered just as a first draft, particularly in its scientific content. Let us hope they will succeed to overcome the  mentioned difficulties in order to propose a revised version deserving more attention. That is not impossible here but more concentration in their endeavor is required for winning the full acceptance of their work in MATERIALS journal.

Author Response

Thank you for reviewing our manuscript. The response of comments and recommendations can be found in the attached file. 

Reviewer 3 Report

In this paper, Mahani Yusoff et al. performed mechanical alloying and sintering method to study the formation of tungsten carbide phases. The authors analysed the solubility of the phases together with the changes of lattice parameter of Cu. Chemical analysis was carried out to describe the atomic binding of W-C interaction. This is a systematic study. However, there are minor issues the author should address before it is suitable for publication.

  1. Authors should describe the bulk structures of WC, WC2, and Cu. Are there any experimental structures of these bulk? If there are, they can provide the figures and lattice parameters in a table.  
  2. Can author comment or discuss on the magnetic and electronic properties of the phase they obtained? Or discussed in the literature?
  3.  Reaction equation 1 can use arrow rather than equal sign.

Author Response

(The authors gave the same response as above.)

Reviewer 4 Report

This paper describes an important topic in materials science. It is well written, but the figures are of poor quality! I definitely recommend to change them all to better resolution, or vector graphics.

On Fig 5. e) and f) subfigures show the EDX spectra. I would recommend to magnify the energy scale, i.e. do not show the spectra above 10 keV, because it is empty there. Larger fonts should be used for the symbols, axes captions, etc. It is unnecessary to use so many digits for the X-ray energy, 1.00 keV, etc. The axes titles are missing, i.e. X-ray energy (keV). There is some axis title on e), but I can’t read it, it is so bad quality, and there isn’t any title and unit on f) vertical axis.

On Fig 9, in each subfigure, there are too many zeros on the vertical axes. Thus it would be better to use kcps units instead of cps, and thus eliminate 3 zeros everywhere.

I spotted several typos, these are easy to fix. I am not sure if I found all of the errors, so I recommend the authors to read the whole text thoroughly again, and correct the typos.

E.g. line 46. 1016, well I suppose this should be 1016 (i.e. 16 should be upper index). In fact the -3 is correcty in upper index position here.

Similarly in line 51. 1015-1016 should be 1015-1016. Here the -3 should be also in upper index, as above in line 46.

Line 77. The -9 should be in upper index, i.e. the pressure is 3x10-9 Torr.

Line 129. 0129 nm, probably it should be 0.129 nm, as 0.138 nm in the line above.

Line 165. W2C: the 2 should be in lower index W2C, similarly to e.g. line 167, or in the abstract line 13. And the same in line 172 and line 175 W2C.

I suppose the final text will be edited such a way to avoid empty space as in lines 292-298, so not only the “Conclusions” title will be there alone at the bottom of the page, but some of the text together with the chapter title.

Paragraph in lines 300-306 should be also justified like the rest of the text, not left aligned.

Author Response

(The authors gave the same response as above.)

Round 2

Reviewer 2 Report

The revised version of the submitted manuscript has been greatly improved compared to the initial one. Just two points mentioned in my short second report are to be considered. 

Author Response

Thank you for your response.

Question 1:

The citations on the effect of milling time on lattice changes of pure metals added line 138-139.  

Question 2:

The sentence for Fe contamination have been added under methodology line 73-74 and  discussion line 209-210

Grammatical error and spelling checked and corrected.